# Virtual Screening for Identification of Dual Inhibitors against CDK4/6 and Aromatase Enzyme

**DOI:** 10.3390/molecules28062490

**Published:** 2023-03-08

**Authors:** Tenzin Adon, Dhivya Shanmugarajan, Hissana Ather, Shaik Mohammad Asif Ansari, Umme Hani, SubbaRao V. Madhunapantula, Yogish Kumar Honnavalli

**Affiliations:** 1Department of Pharmaceutical Chemistry, JSS College of Pharmacy, JSS Academy of Higher Education and Research, Mysuru 570015, Karnataka, India; 2Department of Pharmaceutical Chemistry, College of Pharmacy, King Khalid University, Abha 62529, Saudi Arabia; 3Department of Clinical Pharmacy, College of Pharmacy, King Khalid University, Abha 62529, Saudi Arabia; 4Department of Pharmaceutics, College of Pharmacy, King Khalid University, Abha 62529, Saudi Arabia; 5Department of Biochemistry, JSS Medical College, JSS Academy of Higher Education & Research, Mysore 570015, Karnataka, India; 6Special Interest Group in Cancer Biology and Cancer Stem Cells, JSS Academy of Higher Education & Research, Mysore 570015, Karnataka, India

**Keywords:** dual inhibitor, CDK4/6, aromatase, structure-based pharmacophore, molecular docking, MD simulations, anti-breast cancer drug

## Abstract

CDK4/6 and aromatase are prominent targets for breast cancer drug discovery and are involved in abnormal cell proliferation and growth. Although aromatase inhibitors have proven to be effective (for example exemestane, anastrozole, letrozole), resistance to treatment eventually occurs through the activation of alternative signaling pathways, thus evading the antiproliferative effects of aromatase inhibitors. One of the evasion pathways is Cylin D-CDK4/6-Rb signaling that promotes tumor proliferation and resistance to aromatase inhibitors. There is significant evidence that the sequential inhibition of both proteins provides therapeutic benefits over the inhibition of one target. The basis of this study objective is the identification of molecules that are likely to inhibit both CDK4/6 and aromatase by computational chemistry techniques, which need further biochemical studies to confirm. Initially, a structure-based pharmacophore model was constructed for each target to screen the sc-PDB database. Consequently, pharmacophore screening and molecular docking were performed to evaluate the potential lead candidates that effectively mapped both of the target pharmacophore models. Considering abemaciclib (CDK4/6 inhibitor) and exemestane (aromatase inhibitor) as reference drugs, four potential virtual hit candidates (1, 2, 3, and 4) were selected based on their fit values and binding interaction after screening a sc-PDB database. Further, molecular dynamics simulation studies solidify the stability of the lead candidate complexes. In addition, ADMET and DFT calculations bolster the lead candidates. Hence, these combined computational approaches will provide a better therapeutic potential for developing CDK4/6-aromatase dual inhibitors for HR+ breast cancer therapy.

## 1. Introduction

Breast cancer affects 2.1 million women per year and was the leading cause of cancer-related mortality of 685,000 women globally in 2020 [1,2]. Given that breast cancer continues to be the most prevalent malignancy and the second leading cause of cancer-related death, this constitutes a considerable disease burden [3]. Epidemiological studies have validated that mutations that lead to breast cancer are linked to estrogen levels [4,5]. Estrogen facilitates the estrogen receptor (ER) signaling pathway, which controls several biological processes, including cell proliferation, angiogenesis, and apoptosis [6]. Hormone receptor-positive (HR+) breast cancer harnesses the biological functions of ER pathways to promote breast cancer growth, development, and progression [7]. Therefore, the development of aromatase inhibitors that target the estrogen signaling pathway is one of the key therapeutic approaches for ER+ breast cancer. Despite the introduction of targeted anti-hormonal therapy for HR+ breast cancer, resistance has developed due to ER genome mutations or the activation of other signaling pathways, eventually resulting in relapse [8,9]. Therefore, current clinical practice guidelines recommend the use of anti-hormonal therapy in combination with other novel targeted anti-cancer therapies, which will improve efficacy and reverse hormone resistance [10].

CDK4/6 is a serine–threonine kinase that governs the transition of the cell cycle from the G1 to the S phase. A schematic illustration of the CDK4/6 function is displayed in Figure 1. Deregulation of the CDK4/6-Cylin D-Rb pathway occurs in 90% of breast cancers and is known to be a hallmark of cancer [11]. CDK4/6 inhibitors are most often used in combination with hormonal therapy to increase efficacy and delay disease progression. Such a combination overcomes the efficacy limit of the latter owing to acquired resistance to evade the signaling pathways [12]. Previously, it was reported in patients with HR+/HER2-mBC that the combination of abemaciclib and nonsteroidal aromatase inhibitors resulted in a 50% reduction in the incidence of progression or death [13]. 

The potential benefits of the simultaneous inhibition of CDK4/6 and aromatase were observed in HR+/HER2- breast cancer as an outcome of enhanced significant G1-S cell cycle arrest. This reveals that CDK4/6 inhibitors are complemented by the co-inhibition of aromatase to further increase their therapeutic relevance. As the ER pathway is known to evade through the Cylin D-CDK4/6-Rb pathway to promote tumor growth or proliferation, inhibition of CDK4/6 contributes to aromatase inhibition [14]. The relationship is depicted schematically in Figure 2, further raising the prospect of CDK4/6-aromatase combination therapy. The potential drugs used in treating HR+/HER2- breast cancer with CDK4/6 and aromatase inhibitors in combination therapies are represented in Figure 3.

Given the inherent complications of combinational chemotherapies, such as accumulation of side effects, coordinated dosing of different drugs, the rigor of treatment, undesirable drug interactions, difficulty identifying the cause of side effects, and a considerable increase in financial cost [15], the identification and development of dual CDK4/6-aromatase inhibitors are desirable. Molecules with polypharmacological mechanisms of action are more effective than combination therapy in terms of safety and efficacy, as well as enhanced and simpler administration [16,17]. Computational multitarget drug design techniques enable the development of drugs that can potentially interact with multiple targets simultaneously. One such method is to integrate the pharmacophoric features of the chosen compounds and merge pharmacophore models of multiple targets in virtual screening to find structures that feature fragments that influence two or more targets [18,19].

Hence, this present study’s main objective is to identify the potential lead candidates through various computational chemistry approaches. Thus, the concept of ligand-based drug design utilizing a structure-based pharmacophore approach is the ideal technique to find the common lead candidate. Therefore, structure-based pharmacophore models for CDK4/6 and the aromatase were constructed with statistical validation, followed by mapping, docking, molecular dynamics simulations, ADMET, and DFT, to obtain the potential virtual lead candidates. The models were validated and then put through a pharmacophore-based virtual screening. The potential candidates that satisfactorily fitted both the CDK4/6 and aromatase pharmacophore models were retrieved and docked to these targets. Further, these retrieved candidates were evaluated using in silico pharmacokinetic studies. 

## 2. Results and Discussion

### 2.1. Development of Structure-Based Pharmacophore Model and Validation

Structure-based pharmacophore modeling is a technique for developing pharmacophores based on the active/binding site of the target protein. A pharmacophore is an ensemble of steric and electronic features that contribute to a compound-specific mode of action in the active domain of the targeted macromolecules [20]. The “interaction pharmacophore generation” protocol for each target protein, CDK4/6 (PDB ID: 5L2S) [21] and aromatase (PDB ID: 3S7S) [22], co-crystalized with abemaciclib and exemestane, respectively, are statistically validated internally with receiver operating characteristic (ROC) generated 10 pharmacophore models (Table 1) with excluded volumes, respectively. The validation of pharmacophore models is based on actives and decoys; the detailed information of two drug target details were set (see Appendix A). The best pharmacophore model was selected from each bin of CDK4/6 and aromatase. The pharmacophore_02 model of CDK4/6 comprising six pharmacophore features (AAAADH), represented in Figure 4a, has a good selectivity score of 1.993 and ROC value of 0.909 (Table 1 and Figure 4c). The ROC is an interval validation method of generated pharmacophore models to provide reliable outcomes. To understand the conformation of the ligand binding position, the pharmacophore for neighboring amino acids residues within 5Å was included. As shown in Figure 4b, all pharmacophore features of the pharamcophore_02 model were located around a crucial residue in the CDK6 active site. The hydrogen bond acceptor (A1) is located near the water molecule HOH1011. The second hydrogen bond acceptor (A2) is near the α-NH_2_ group of the Val101 residue. The third and fourth hydrogen bond acceptors (A3 and A4) were found near the ε-NH_2_ group of Lys43. The hydrogen bond donor (D) is located near the C=O group of Val101. The hydrophobic feature (H) is located on the heterocyclic fused ring of abemaciclib, pointing towards Ile19 and Val27.

Similarly, the pharmacophore_09 model of aromatase consisting of four pharmacophore features (AHHH), represented in Figure 4d, with a high selectivity score of 6.347, was chosen based on the highest acceptable ROC value of 0.815 (Table 1 and Figure 4f). As displayed in Figure 4e, four pharmacophore features include one hydrogen bond acceptor (A) towards an α-NH_2_ group of Met374, and the remaining three hydrophobic features (H1, H2, and H3), which are near to each other, pointed towards Phe134, Tyr224, and Val370 amino acid residues at the binding site of aromatase. Consequently, the sc-PDB database [23] “http://bioinfo-pharma.u-strasbg.fr/scPDB/ (accessed on 12 January 2022)”was used for screening the two structure-based pharmacophore models of CDK6 and aromatase.

### 2.2. Pharmacophore-Based Virtual Screening

The pharmacophore-based virtual screening approach is one of the cost-effective techniques in ligand-based drug design to identify potential lead candidates [24]. Moreover, this technique aids researchers in selecting better compounds for further analysis. CDK4/6 and aromatase inhibitors pharmacophore models (see Appendix A, respectively) were screened within the built-in sc-PDB database “http://bioinfo-pharma.u-strasbg.fr/scPDB/ (accessed on 12 January 2022)” of DS v.2019 consisting of diverse chemical compounds, with a count of approximately 7K. 

A total of 755 compounds were obtained based on fit value, 37 from the pharmacophore_02 model of CDK4/6 and 718 from the pharmacophore_09 model of aromatase inhibitors. The highest fit value of the compound will map with the maximum pharmacophoric features. 

Among 755 compounds, only four compounds (Figure 5a) were identified in integration or common with two pharmacophore models with fit values ranging from 0.748 to 3.538 (Table 2). The molecular structures of the identified potential virtual hit and their mapping against both pharmacophore models are shown in Figure 5. Candidate 2 has been previously reported for aromatase and steroid sulfatase inhibition [25]. This solidifies that the pharmacophore-based virtual screening method is an effective tool in drug discovery for identifying potential compounds with desired biological activity. Lead candidate 2 reported for aromatase inhibitory activity shares its structural similarity with the drug letrozole, a known aromatase inhibitor, while its reported steroid sulfatase inhibitory function enhances its aromatase inhibitory effects.

The other lead candidates 3 and 4 possessing tetrahydroisoquinoline scaffolding were reported as moderate carbonic anhydrase I inhibitors with inhibition constants (Ki) of 2.8 µM and 6.41 µM, respectively [26]. Tetrahydroisoquinoline scaffold-consisting compounds were known to exhibit CDK4 [27] and aromatase [28] inhibition. Thus, lead candidates 3 and 4 are structurally similar and may also exhibit inhibition of CDK4/6 and aromatase.

### 2.3. Molecular Docking Analysis of Identified Potential Candidates

The binding interaction energy of the screened lead candidates with standard compounds (abemaciclib and exemestane) was docked with respective drug targets CDK6 (PDB ID:5L2S) and aromatase (PDB ID:3S7S). Validation of the CDOCKER molecular docking protocol was carried out to obtain the native-like conformation during sampling [29]. The optimized parameters of CDOCKER have the ability to generate RMSD values of 1.821 Å (abemaciclib) and 1.139 Å (exemestane), respectively. The molecular docking results of lead candidates against CDK6 demonstrated moderate CDOCKER energy and CDOCKER interaction energy compared with the standard drug abemaciclib. However, docking of lead candidates at the active site of aromatase depicts better interaction energies compared to the standard drug, exemestane (Table 3). This result shows the promising binding of lead candidates against both the target proteins CDK6 and aromatase. These findings of the docking study and structure-based pharmacophore virtual screening of lead candidates bolster the idea that the screened lead candidates might concurrently inhibit CDK4/6 and aromatase. 

### 2.4. Binding Modes Analysis of Identified Candidates

The co-crystal ligands abemaciclib (PDB ID:5L2S) and exemestane (PDB ID:3S7S) showed potent inhibitory activity against CDK4/6 and aromatase proteins, respectively. To examine the binding modes of the identified candidates with the target proteins, CDK6 and aromatase, the co-crystal ligands served as references. The abemaciclib binds to the active binding site of CDK6 with a CDOCKER energy of −27.35 kcal/mol and a CDOCKER interaction energy of −58.02 kcal/mol. Abemaciclib interacted with the key residue of CDK6, including hydrogen bond interactions with Val101 and Lys43 and hydrophobic interactions with Ile19, Tyr24, Val27, Ala41, and Leu152 (Figure 6a).

Exemestane binds to the active binding site of aromatase with a CDOCKER energy of −22.01 kcal/mol and a CDOCKER interaction energy of −39.15 kcal/mol. Exemestane demonstrated molecular interactions at the active site of the aromatase (PDB ID:3S7S) with key residues, including hydrogen bonding interactions with Arg115 and Met374 and hydrophobic interactions with Phe134, Phe221, Trp224, Ala306, and Val370 (Figure 6b).

#### 2.4.1. Binding Mode of Candidate 1

The candidate 1 potential modes of binding to CDK6 and aromatase active sites have been elucidated. For CDK6, this candidate 1 has CDOCKER energy of −13.04 kcal/mol, CDOCKER interaction energy of −56.37 kcal/mol (Table 3), and demonstrated interactions with amino acid residues such as Ile19, Tyr24, Lys147, and Asp163 (Figure 7a). A thorough analysis of the binding interaction of candidate 1 in the catalytic site of CDK6 revealed that four conventional hydrogen bonds were formed. The carbonyl group and adjacent hydroxyl group on the oxane ring of candidate 1 formed the first and second hydrogen bonds with the εNH3 group of Lys147. Meanwhile, the third and fourth hydrogen bond is formed between the candidate 1 hydroxyl groups on the oxane ring and the Asp163 γ-carbonyl group. The study reported that this kind of interaction with the backbone of Asp163, a DFG motif of the activation segment, contributes to a potent and selective CDK4/6 profile. The attractive charge between the candidate 1 carbonyl group and the εNH3 group of Lys147 was observed. Candidate 1 also forms hydrophobic alkyl and pi–alkyl interactions with Ile19. Try24 of CDK6 forms a non-classical hydrogen bond known as a carbon–hydrogen bond with the oxane ring of candidate 1. Furthermore, close contact van der Waals interactions between the candidate 1 and CDK6 residues, including key residues, such as Val27, Ala41, Lys43, and leu152, were observed. Amino acid residues were removed for clarity. The observed van der Waals interaction with His100 has been reported to be an important determinant of selectivity over other kinases.

For aromatase, this candidate 1 has scored a CDOCKER energy of 3.892 kcal/mol, CDOCKER interaction energy of −56.27 kcal/mol, and has shown interactions with residues including Arg115, Ile133, Met303, Ala306, Asp309, Val370, Leu372, Met374, Cys437, and Leu477 (Figure 7b). The presence of three hydroxyl groups on the oxane ring moiety of candidate 1 established four hydrogen bonds, two with the NH group of Met374 and the remaining two with the carbonyl group of Leu372 and Leu77. The Arg115 NH group also forms hydrogen bond interactions with the O-18 atom of candidate 1 located between the oxane and phenyl ring systems. The benzene rings of candidate 1 formed hydrophobic pi–alkyl interactions with Ile133, Ala306, and Val370 at the aromatase active site, while pi–sulfur interaction with Met303 and Cys437 is observed. Several residues, including Phe134, Phe221, Trp224, Thr310, Val373, and Ser478, are involved in van der Waals interactions. Additionally, non-classical hydrogen bonding between the, a methyl group, and the carbon atom of the oxane ring of candidate 1, with the, the carbonyl group of Asp309, and Leu372, of aromatase are revealed.

#### 2.4.2. Binding Mode of Candidate 2

Candidate 2 demonstrated a CDOCKER energy of −18.21 kcal/mol and a CDOCKER interaction energy of −46.60 kcal/mol (Table 3). Docking at the catalytic domain of CDK6 revealed significant interactions between candidate 2 and key residues Ile19, Val101, Asp104, Leu152, and Asp163 (Figure 8a). Additionally, interactions with Glu21, Try24, Lys147, Gln149, and Phe172 were observed. A crucial hydrophobic interaction is established between one of the benzene ring moieties of candidate2 with Ile19 and Leu152 and between the methoxy carbon residing near Val101 and Leu152. Additionally, a hydrophobic pi–alkyl interaction between the second methoxy carbon of candidate 2 and Try24. A total of five hydrogen bond interactions was observed, including important hydrogen bond interactions between the key amino acid residue Val101 and the NH_2_ group of candidate 2. The remaining hydrogen bond was observed between the methoxy and sulfonyl groups of candidate 2 with the εNH_3_^+^ group of Lys147, the NH2 group with the carbonyl group of Glu21, and the oxy atom of the phenyl-sulfamate moiety with the OH group of Tyr24. Furthermore, non-classical hydrogen bonds were observed between the candidate 2 triazole ring with CDK 6 Asp104 and each methoxy group with Val101 and Lys147 residues, respectively. Other interactions such as pi–sulfur between the sulfamate of candidate 2 and Phe172 and van der Waals interactions with residues, including important residues Ala41, Val27, Glu99, His100, and HOH1011, were observed.

Regarding molecular docking of candidate 2 at the active site of aromatase, it exhibited a CDOCKER energy of −25.29 kcal/mol and CDOCKER interaction energy of −52.21 kcal/mol (Table 3). The elucidation of the potential binding modes of candidate 2 in the aromatase active domain demonstrated several interactions with amino acid residues such as Arg115, Ile133, Ile132, Trp141, Arg145, Ala306, Asp309, Val370, Val373, and Cys437 (Figure 8b). The methoxy and sulfonyl amide groups of candidate 2 formed conventional hydrogen bond interactions with key aromatase residues such as Arg115, Ile132, Trp141, and Arg145. Moreover, the phenyl ring of Trp141 is also involved in pi–sulfur interaction with the sulfonyl amide moiety of the candidate 2 compounds. The aromatase- candidate 2 complexes are also supported by several hydrophobic alkyls and pi–alkyl interactions between Ile133, Ala306, Val370, Val373, and Cys437 of aromatase and the methyl, phenyl, and triazole ring system of candidate 2. A non-classical hydrogen bond was observed between the sulfonyl group of the candidate 2 compound and the aromatase Arg435 residues. Furthermore, the complex is supported by van der Waals interactions with the key residues Phe221, Trp224, Thr310, Ser478, and leu477, among other residues.

#### 2.4.3. Binding Mode of Candidate 3

The elucidation of CDK6-candidate 3 complex interactions with CDOCKER energy of −31.82 kcal/mol and CDOCKER interaction energy −35.28 kcal/mol (Table 3) was supported by several interactions with key amino acid residues such as Ile19, Val27, Lys43, Val101, Asp104, Leu152, and Ala162 (Figure 9a). The sulfonyl group of the piperidine−1-sulfonamide ring system forms a hydrogen bond with the εNH_3_⁺ group of Lys43. Both methoxy groups were involved in non-classical hydrogen bond interactions with Val101 and Asp104. Significant hydrophobic interactions with Ile19, Val27, Leu152, and Ala162 in the active-site region were observed. Moreover, the δ2 -carbon of Leu152 also formed a pi–sigma interaction with the benzene group of candidate 3. Additionally, van der Waals interactions occur between the candidate 3 and CDK6 active site residues, including Gly20, Try24, Hisp100, Asp102, Gln103, Gln149, Asn150, and Asp163.

The molecular docking interaction between the aromatase and candidate 3 complex is supported by several types of interactions, including conventional and non-classical hydrogen bonding, hydrophobic alkyl, and pi–alkyl interactions, and van der Waals contacts with CDOCKER energy of −28.57 kcal/mol and CDOCKER interaction energy of −32.98 kcal/mol (Table 3). The Cys437 NH and Ala306 carbonyl groups are involved in conventional and non-classical hydrogen bonding with the sulfonyl and methoxy groups of candidate 3, respectively (Figure 9b). Hydrophobic alkyl and pi–alkyl interactions with important residues Ile133, Trp224, Leu477, and Val370 were demonstrated. Furthermore, van der Waals interactions with several residues were observed, especially with key residues, such as Arg115, Phe221, Asp309, Thr310, Leu372, and Ser478. 

#### 2.4.4. Binding Mode of Candidate 4

The candidate 4 compound has scored the CDOCKER energy of −30.69 kcal/mol and the CDOCKER interaction energy of −32.07 kcal/mol against the CDK6 (Table 3). In terms of CDK6, candidate 4 demonstrated various kinds of interactions, such as alkyl, pi–alkyl, van der Waals, water–hydrogen bond, and non-classical and conventional hydrogen bonding interactions with the amino acid residues in the active domain of CDK6 (Figure 10a). The sulfonyl and NH_2_ groups of the piperidine−1-sulfonamide ring of candidate 4 engage in crucial conventional hydrogen bonding interactions with the amine and carbonyl groups of Val101, respectively. Additionally, a hydrogen bond interaction between NH_2_ of candidate 4 and the carbonyl group of CDK6 residue Gln103 was observed. The methoxy group was involved in non-classical hydrogen bonding with the hydroxyl group of Asp163. Moreover, the methoxy carbon atom of candidate 4 also formed a pi–alkyl interaction with the phenyl ring of the Try24 residues of CDK6. 

The C1 of 3,4-dihydroisoquinoline-2(1*H*)-sulfonamide formed notable hydrophobic interactions, such as alkyl interactions, with Leu152 and Ala162 in the active region. The sulfonyl group of 3,4-dihydroisoquinoline-2(1*H*)-sulfonamide of candidate 4 forms a water–hydrogen bond with HOH1011 within the active site of CDK6. The other amino acid residues, such as Ile19, Val27, His100, Asp102, Asp104, Gln149, and Asn150, were involved in van der Waals interactions. 

The molecular docking interaction between the aromatase and candidate 4 complex is represented in Figure 10b. The sulfonyl group of candidate 4 established a hydrogen bond interaction with Arg115 in the active site of aromatase. This key interaction plays a significant role in the potency and selectivity of the aromatase inhibitors. Moreover, the 2nd sulfonyl oxygen atom was bonded with the Cys437 αH atom through non-classical hydrogen bond interactions. The 1,2-dimethoxybenzene ring formed pi–pi interactions with Trp224 and pi–alkyl interactions with Val370 and Ile133. However, its methoxy substituent group formed an alkyl interaction with Leu477. The piperidine ring of candidate 4 interacts with Ile133 through alkyl interactions and van der Waals contacts with aromatase active sites, including residues Phe134, Phe221, Asp309, Thr310, and Ser478. The CDOCKER energy of −30.35 kcal/mol and the CDOCKER interaction energy of −31.16 kcal/mol were exhibited by candidate 4 at the catalytic domain of aromatase (Table 3). In summation, these four identified candidates demonstrated molecular interactions akin to the key interactions of clinically known inhibitors of CDK6 and aromatase at the active domains of the target proteins.

### 2.5. Molecular Dynamic Analysis of the Promising Candidates

The identified virtual candidates were put through MD simulations to verify the outputs of molecular docking and evaluate the stability of docked compounds in both the binding pockets of CDK6 (PDB ID 5L2S) and aromatase (PDB ID 3S7S) proteins. All systems were employed for a 50 ns time scale MD simulation and calculated root-mean-square deviation (RMSD), root-mean-square fluctuation (RMSF), and protein–ligand contact interaction using MD trajectories. As shown in Figure 11a, for the CDK6- candidate1 complex, the RMSD value of the protein backbone rose to about 1.8 Å during the first 15 ns and remained stable at around 2.2 Å until the simulation ended. Furthermore, the RMSD value of the candidate 1 configuration fluctuated frequently in the first 8 ns and reached equilibration at approximately 2 Å after 10 ns during the entire simulation. The same ligand complexes with protein aromatase (PDB ID: 3S7S) were observed to remain stable during the course of the MD simulation within the order of 3 Å (Figure 12a). These results indicated that candidate 1 attained a stable state with both CDK6 and aromatase proteins at the end of the 50 ns run time. The protein’s alpha carbon in the CDK6- candidate 2 complexes also displayed stable variation with an RMSD of less than 2.1 Å, while protein-bound ligand (candidate 2) configuration fluctuated between 0.9 Å to 2.2 Å at the first 43 ns and remained stable around 1.8 Å after 45 ns of the MD simulation (Figure 11b). The same ligand in complex with aromatase displayed acceptable protein and ligand RMSD and reached maximum deviations of 2.1 and 1.2 Å RMSD, respectively, at intervals of 50 ns (Figure 12b).

The CDK6- candidate 3 complex simulation trajectories indicated an equilibrium and stable RMSD for the protein throughout a 50 ns timeframe. Within 40 ns, the protein-bound candidate 3 had acceptable maximum deviations of 3.2 Å RMSD, followed by a consistent trajectory with a maximum deviation of 5.2 Å until 50 ns of the simulation (Figure 11c). The same ligand in complex with aromatase exhibited acceptable protein and ligand RMSD and reached maximum deviations of 2.2 and 5.6 Å RMSD, respectively, at intervals of 50 ns (Figure 12c). The RMSD of the protein alpha carbon atoms in the CDK6-candidate4 complex was found to be between 1.6 and 2.6 Å, while RMSD of protein-bound ligand candidate 4 fluctuated between 1.2 and 2.5 Å from 0 to 42 ns, then stabilized at 0.8 Å to 2.0 Å until 50 ns (Figure 11d). For the aromatase–candidate 4 complex, both protein and protein-bound ligand RMSD fluctuate at the first 10 ns and attained equilibration at 1.75 and 5.6 Å, respectively, after 10 ns of MD simulation (Figure 12d).

The corresponding RMSF study further corroborated the RMSD deviations in the protein and ligand (see Appendix A), where the heavy ligand atoms and protein residues both showed acceptable oscillations. Furthermore, protein–ligand interaction profiling was undertaken on each stimulated complex, and the 50 ns simulation trajectories indicated the establishment of hydrogen bonds, and hydrophobic, ionic, and water bridges as a function of time (see Appendix A). Throughout the MD simulations, these protein–ligand intermolecular interactions determined the binding affinity and stability of candidates at CDK6 and aromatase catalytic sites. At the end of the simulation period, the ultimate poses for the interacting protein–ligand complexes were obtained (see Appendix A), further supporting the ligand residence in the catalytic domain of both CDK6 and aromatase protein complexes.

### 2.6. Predicted ADMET Analysis

ADMET (adsorption, distribution, metabolism, excretion, and toxicity) aspects are important during the drug development process since they account for 60% of drug molecule failure [30]. These properties primarily influence drug bioavailability, cell permeability, and metabolism, which are vital parameters in drug discovery research. As a result, the candidates were subjected to ADMET property prediction utilizing the DS client and the software toxicity prediction wizard [31]. Table 4 presents the expected ADMET values. TOPKAT AMES mutagenicity projected that all potential candidates were non-mutagenic. Additionally, the ADMET plot was constructed using the calculated PSA_2D vs. AlogP98, illustrated in Figure 13. The binary PSA_2D vs. AlogP98 plot indicates the 99% and 95% confidence ellipses corresponding to human intestinal absorption (HIA) and blood–brain barrier (BBB) penetration models, predicting the cellular permeability of compounds [32]. According to the model, the compound should meet the following specifications to achieve optimal cellular permeability: PSA_2D < 140 Å^2^ and AlogP98 < 5 [32]. All the hit candidates except for candidate 2 showed PSA_2D < 140 Å^2^, suggesting good cell permeability. Therefore, candidates 1, 3, and 4 are present within the 99% confidence ellipse regarding absorption, while candidate 2 fails, indicating that the former-mentioned candidates are expected to have good HIA. All candidates are predicted to exhibit poor BBB permeability (levels 3 and 4, as shown in Table 4). The oral bioavailability of drugs is influenced by their aqueous solubilities. Except for candidate 2 (level 2), all the identified candidates have been predicted to have acceptable aqueous solubility. The cytochrome P450 (CYP2D6) isoenzyme is implicated in the metabolism of a wide range of drugs, and inhibiting it with a drug can result in fatal drug–drug interactions [33]. None of the candidate compounds demonstrated any CYP2D6 inhibitory action, as can be seen in Table 4. Several variables influence drug distribution from the plasma to target tissues, with binding to plasma proteins being (PPB) the most critical [34]. According to this prediction, candidate 2 is unlikely to bind to serum carrier proteins, but the remaining candidates indicate plasma PPB. All the candidates identified have the potential to be hepatotoxic.

### 2.7. Molecular Orbital Properties

The energies of the highest occupied molecular orbital (HOMO) and lowest unoccupied molecular orbital (LUMO) of the potential candidates are listed in Table 5. The energy gap (ΔE) between the HOMO and LUMO is associated with the chemical reactivity of the molecule [35,36]. A molecule is more reactive if it has a lower ΔE. The possible charge–transfer interaction that occurs within the molecules is explained by the lower value of the ΔE [37]. In this study, the results indicate that the trend in the ΔE for the candidate compounds is candidate 1 < candidate 4 < candidate 3 < candidate 2. Candidate 1 has the lowest ΔE, indicating that it is the least stable and more reactive than the others.

## 3. Materials and Methods

### 3.1. Preparation of Proteins

The 3D structure of CDK6 (PDB ID:5L2S) and aromatase (PDB ID:3S7S) having a resolution of 2.27 Å and 3.21 Å has been retrieved from the RCSB protein data bank “https://www.rcsb.org/ (accessed on 10 January 2022)”. Proteins were prepared using the “prepare protein wizard” of Discovery Studio 2019 (DS.v.2019) [38]. Missing hydrogen atoms, residues, side chains, and total charges were generated. The protein was then optimized and energy-minimized using the CHARMM force field to achieve the structural integrity of the final proteins.

### 3.2. Generation and Validation of Structure-Based Pharmacophore Models

The selected proteins of interest (PDB ID:5L2S and 3S7S) co-crystallized with active ligands (abemaciclib and exemestane, respectively) were analyzed. Using the “interaction pharmacophore generation” wizard of DS.v.2019 [38], energetically optimized structure-based pharmacophore models were generated based on the complementary interactions between the protein and ligand. Biovia Discovery Studio software employs a genetic function algorithm (GFA) to rank the generated pharmacophore models. Internal validation of the generated pharmacophore models was then performed using the receiver operating characteristic (ROC) curve and area under the ROC curve (AUC), providing structurally validated pharmacophore models. Therefore, 17 and 16 experimentally active compounds against CDK4/6 and aromatase, respectively, were retrieved from the ZINC database as active decoys. Inactive decoy sets of 639 and 289 of CDK4/6 and aromatase, respectively, were retrieved from the Directory of Useful Decoys enhanced (DUD-E) “http://dude.docking.org/ (accessed on 10 January 2022)”. This validation examined how effectively these pharmacophore models could differentiate between the active and inactive molecules.

### 3.3. Pharmacophore-Based Virtual Screening

The best-validated pharmacophore models of CDK4/6 and aromatase, respectively, were used to filter the sc-PDB database “http://bioinfo-pharma.u-strasbg.fr/scPDB/ (accessed on 12 January 2022)” by the “screen library” module of DS.v. 2019 [38] to probe possible dual inhibitors against these two target proteins. Upon screening the databases, molecules were ranked according to their fitness values. The higher the fit value of a particular molecule, the better the molecule fits into the pharmacophore model. Based on the observed capability to successfully map both target receptor pharmacophore models, the dual inhibitory properties with the highest fit values were identified and selected as possible dual candidates against CDK4/6 and aromatase.

### 3.4. Molecular Docking

Molecular docking was performed to generate the bioactive binding conformation of possible candidates, along with the reference ligands abemaciclib and exemestane. The CDOCKER protocol based on simulation annealing and CHARMm in DS.v.2019 was used to perform molecular docking [29] and validated in terms of RMSD to ensure the accuracy of the docking protocol. RMSD was calculated by comparing two poses, that is, co-crystal poses and redocked poses of the same ligand. To accurately reproduce the experimentally reported binding mode of the ligands, the CDOCKER protocol must have an RMSD value of less than 2.5 Å. Following molecular docking, the candidates were ranked based on the scoring function of the CDOCKER protocol, including CDOCKER energy and CDOCKER interaction energy. The target protein and ligand bind more effectively when the negative CDOCKER energy is high. Discovery Studio Visualizer 2019 was used to investigate the protein–ligand interaction complexes.

### 3.5. Molecular Dynamics (MD) Simulation

The Schrodinger suite software’s Desmond module was employed for MD simulation to investigate the atomic-level binding stability and consistency of the candidates [39,40]. The complexes of candidate 1–4 with CDK6 (PDB ID: 5L2S) and aromatase (PDB ID: 3S7S) were solvated in an orthorhombic box with a buffer region of 10 Å between the protein atoms and box edges using the explicit TIP3P water model. To neutralize the charges, enough counter ions such as chloride ions were introduced, followed by the addition of 0.15 M NaCl salt concentration to imitate human physiological conditions.

The constructed system was then subjected to energy minimization using preset OPLS3e force field parameters to accurately align the protein structure within the simulation constraints and to minimize electronic conflicts between protein structures [41,42]. The energy-minimized system was then put through a 50 ns MD simulation with an “isothermal-isobaric ensemble” (NPT) employing a 1 bar using Martyna–Tobias–Klein barostats and a Nose–Hoover thermostat at 300 K [43]. Every 1000 ps, the simulation trajectory was recorded, and the resulting trajectories were examined to assess the stability of the complexes in terms of RMSD, RMSF, and protein–ligand interactions. RMSD assesses the stability of the protein–ligand complex by analyzing the equilibration period of the MD run and denoting the dynamic changes in the protein and ligand at specific temperatures and pressures over the simulation time [44]. During isothermal and isobaric processes, the protein residue fluctuation of a protein–ligand complex is determined by RMSF [45]. The stability of protein–ligand binding increases with decreasing fluctuation range or protein residue involvement, increasing the likelihood that the ligand will remain at the protein-binding site for the duration of the MD simulation run.

### 3.6. ADMET Prediction and Calculation of Molecular Orbitals

Furthermore, we performed in silico prediction of adsorption, distribution, metabolism, excretion, and toxicity (ADMET) parameters for the identified candidates using ADMET descriptors and toxicity prediction tools from DS.v.2019 [33]. Defined ADMET descriptors, such as HIA, calculated at aqueous solubility at 25 °C, BBB penetration after oral administration, PPB, CYP2D6 inhibition, and hepatotoxicity, were evaluated for the selected candidates and are listed in Table 4. This broadens the possibilities for future drug-molecule development. The results are interpreted based on the graphical representation plotted against ADMET_AlogP98 vs. ADMET_PSA 2D in the optimum prediction space, which shows a confined level of 95% and 99% for BBB and human intestinal absorption, respectively. The identified candidates were analyzed for TOPKAT AMES mutagenicity. The level of interpretation of the results for ADMET and TOPKAT is tabulated in Table 6. 

### 3.7. Density Functional Theory (DFT) Calculations

Calculations based on DFT were utilized to determine a compound’s primary molecular properties. In this inquiry, DFT calculations were used to optimize the 3D structures of the four identified candidates [46]. These structures were subsequently used in the computations of the molecular orbitals, that is, the highest occupied molecular orbital (HOMO), lowest unoccupied molecular orbital (LUMO), and energy gap (ΔE).

## 4. Conclusions

CDK4/6 and aromatase are validated targets for the treatment of breast cancer, which is currently one of the most frequently diagnosed cancers. The simultaneous inhibition of CDK4/6 and aromatase has established a suitable strategy to develop a dual-enzyme inhibitor based on the evidence of therapeutic benefits over inhibiting one or the other target. To date, no computational method has been developed for identifying dual inhibitors of CDK4/6 and aromatase. With this background, we proposed dual CDK4/6 and aromatase inhibitors, which can be further evaluated by in vitro and in vivo biological tests. To achieve the design of dual inhibitors, a virtual screening workflow was applied, including the generation of pharmacophore models, followed by pharmacophore-based virtual screening of seven thousand compounds from the sc-PDB database and molecular docking studies of identified candidates, MD simulation, and ADMET analysis. The results showed that there were four candidates (1, 2, 3, and 4, Figure 5a) that mapped well onto features of our selected pharmacophore models and had an optimum docking score compared to abemaciclib and exemestane as reference drugs. Further, these candidates replicated almost similar interactions as reference drugs and exhibited favorable pharmacokinetic properties, except for candidate 2, and were predicted to be non-mutagenic. Pharmacophore-based screening approaches not only facilitated the elimination of compounds that lacked the fundamental chemical features required for dual inhibition of the target proteins, but they also enabled us to avoid false positives during docking studies beforehand. Molecular dynamics simulation investigations have demonstrated the compactness and stability of identified candidates (1, 2, 3, and 4, Figure 5a) in the binding sites of both protein targets. Furthermore, the compounds having a tetrahydroisoquinoline scaffold, as seen in candidates 3 and 4, have been demonstrated to inhibit CDK4 and aromatase. Thus, based on these observations, potential candidates 1, 2, 3, and 4 could be possible dual inhibitors of CDK4/6 and aromatase. Moreover, in vitro and in vivo activity of synthesized derivatives of candidate 4 as CDK4/6-aromatase dual inhibitors is currently underway and will be reported soon.

## Figures and Tables

**Figure 1 molecules-28-02490-f001:**
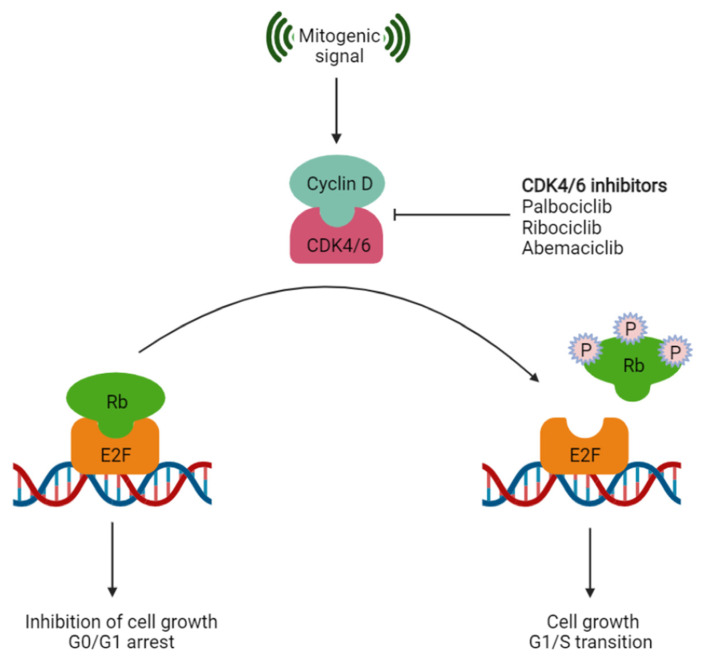
A schematic diagram of CDK4/6 function depicting the role of phosphorylation and inactivation of retinoblastoma protein (Rb). The mitogenic signal stimulates the formation of the Cyclin D-CDK4/6 complex, which subsequently phosphorylates Rb, leading to the release of E2F transcription factors, thus triggering the expression of the gene responsible for the G1-S phase transition. CDK4/6 inhibitors prevent Rb phosphorylation, thereby arresting the cell cycle.

**Figure 2 molecules-28-02490-f002:**
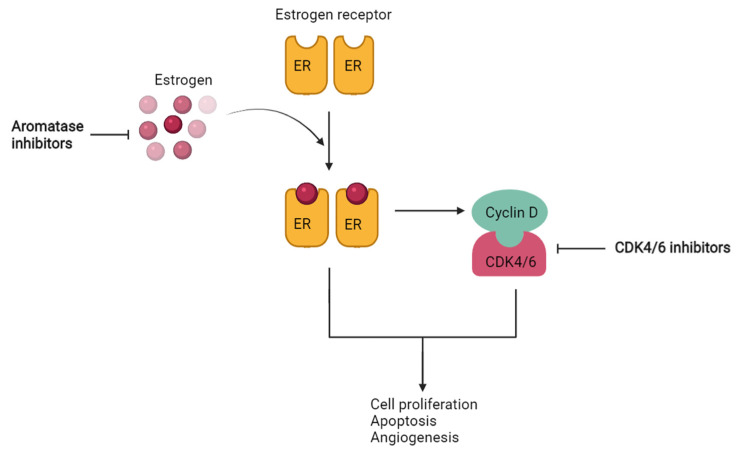
A schematic illustration of the functional relationships between aromatase and CDK4/6 inhibitors. The ER pathway is directly targeted by aromatase inhibitors, while CDK4/6 inhibitors work by inhibiting the ER escape pathway, or the Cyclin D-CDK4/6-Rb pathway.

**Figure 3 molecules-28-02490-f003:**
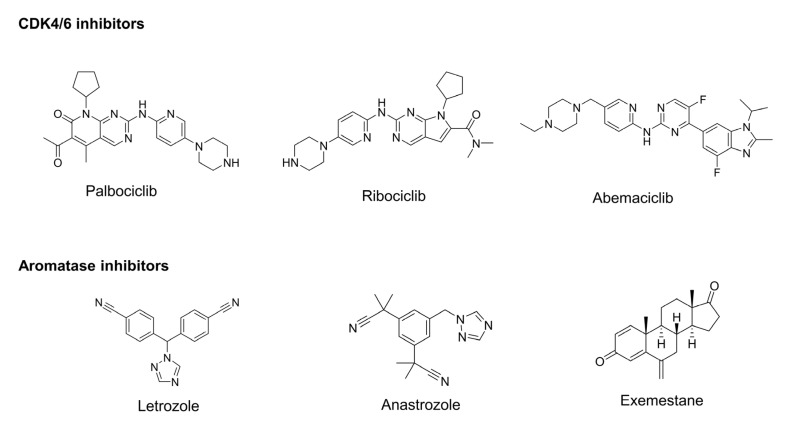
Representative CDK4/6 and aromatase inhibitors have been studied for their potential in combined therapy.

**Figure 4 molecules-28-02490-f004:**
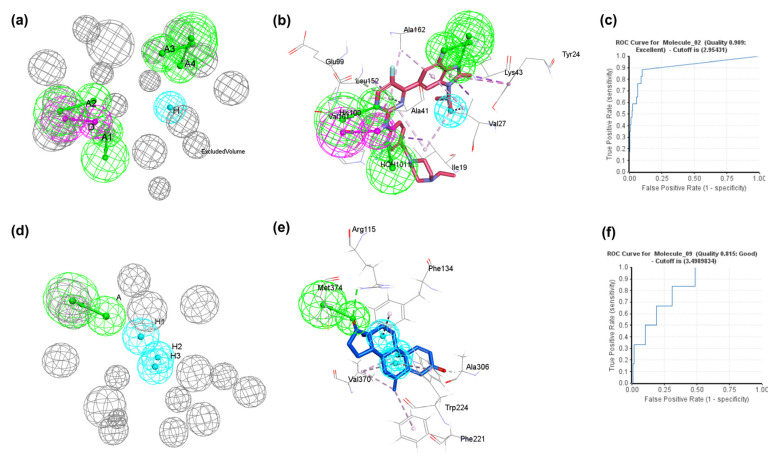
Representation of pharmacophore models. (**a**) Pharmacophore _02model (AAAADH) of CDK6 protein with six pharmacophoric features; (**b**) pharmacophore_02 model in complex with co-crystal abemaciclib (PDB ID:5L2S); (**c**) ROC curve of the validation of pharmacophore_02 model; (**d**) pharmacophore_09 model (AHHH) of aromatase with four pharmacophoric features; (**e**) pharmacophore_02 model in complex with co-crystal exemestane (PDB ID:3S7S); (**f**) The validation ROC curve for the pharmacophore_09 model. Magenta, green, and cyan represent hydrogen bond donor (D), hydrogen bond acceptor (A), and hydrophobic (H) features, respectively. The grey color represents the excluded volumes.

**Figure 5 molecules-28-02490-f005:**
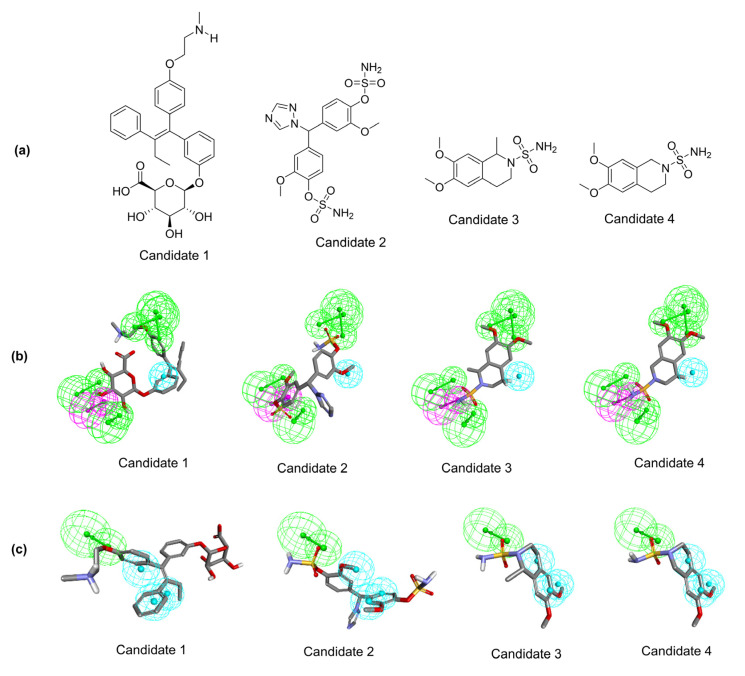
The structure-based pharmacophore model identified candidates; (**a**) 2D structure of identified virtual hit candidates, (**b**) pharmacophore_02 model (AAAADH) aligns to the identified hit candidates, (**c**) pharmacophore_09 model (AHHH) aligns to the identified virtual hit candidates.

**Figure 6 molecules-28-02490-f006:**
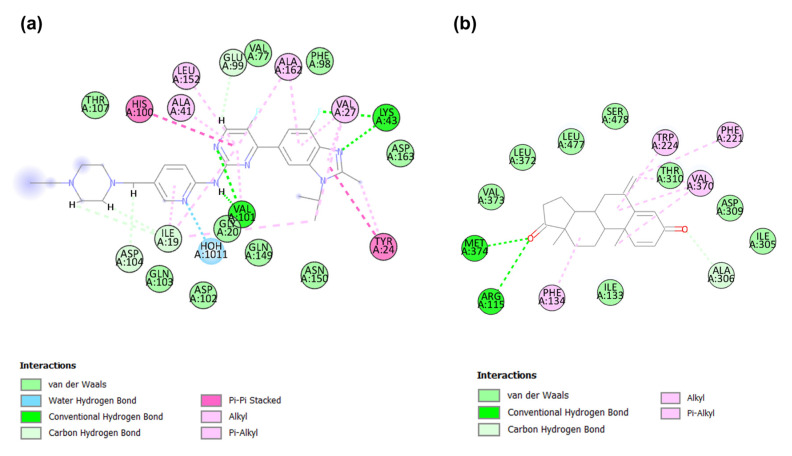
(**a**) shows the 2D docking interactions between abemaciclib and CDK6 (PDB ID: 5L2S). (**b**) shows the 2D interactions between exemestane and aromatase (PDB ID: 3S7S).

**Figure 7 molecules-28-02490-f007:**
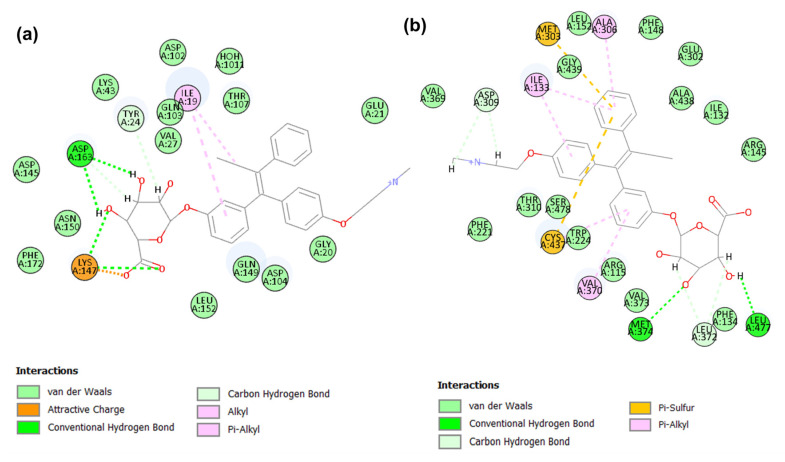
The 2D plot of candidate 1 in CDK6 (**a**) and aromatase (**b**) at the binding site.

**Figure 8 molecules-28-02490-f008:**
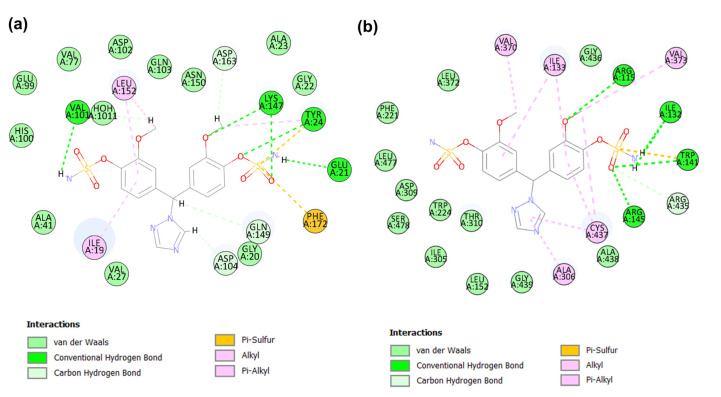
2D plot of potential candidate 2 in CDK6 (**a**) and aromatase (**b**) at the binding site.

**Figure 9 molecules-28-02490-f009:**
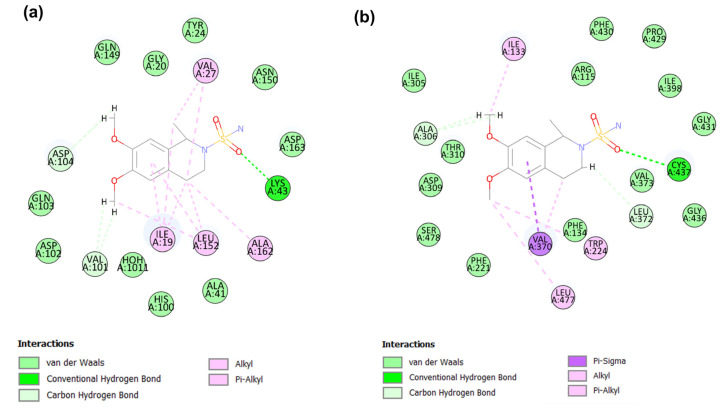
The 2D plot of candidate 3 in CDK (**a**) and aromatase (**b**) at the binding site.

**Figure 10 molecules-28-02490-f010:**
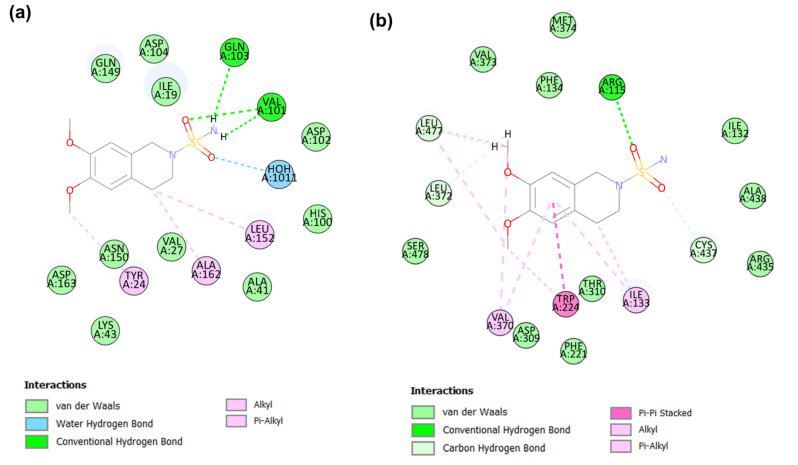
The 2D plot of candidate 4 in CDK (**a**) and aromatase (**b**) at the binding site.

**Figure 11 molecules-28-02490-f011:**
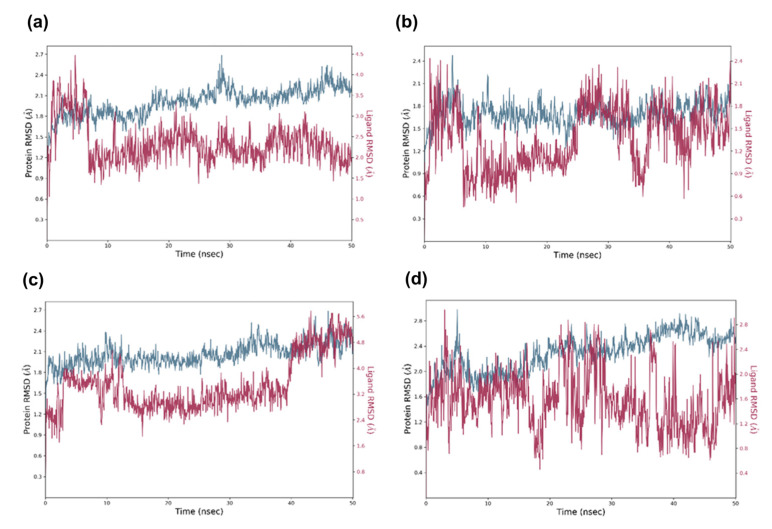
MD simulation trajectory (50 ns) analysis of the docked complexes via RMSD. In the time-dependent RMSD graph, Cα-atoms of CDK6 complex with identified potential candidates, i.e., (**a**) candidate 1, (**b**) candidate 2, (**c**) candidate 3, (**d**) candidate 4. The CDK Cα-atom RMSD is shown in blue, while the RMSD of candidates concerning CDK6 is shown in red.

**Figure 12 molecules-28-02490-f012:**
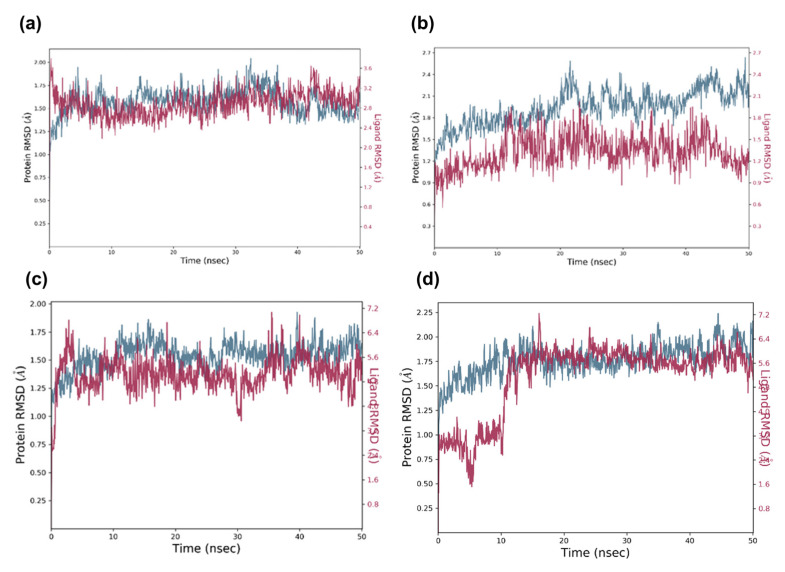
MD simulation trajectory (50 ns) analysis of the potential docked complexes via RMSD. In the time-dependent RMSD graph, Cα-atoms of aromatase complex with identified potential virtual candidates, i.e., (**a**) candidate 1, (**b**) candidate 2, (**c**) candidate 3, (**d**) candidate 4. Aromatase Cα-atom RMSD is shown in blue, whereas the RMSD of candidates for aromatase is shown in red.

**Figure 13 molecules-28-02490-f013:**
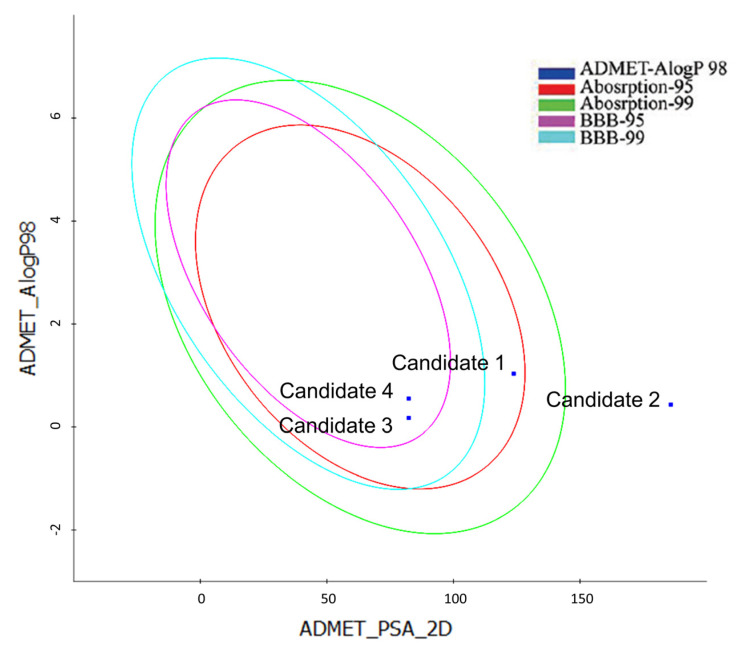
ADMET PSA vs. ALogP98 plot for identified compounds demonstrating the 95% and 99% confidence limit ellipses corresponding to the BBB and HIA models. According to the model, the compound should meet the following specifications to achieve optimal cellular permeability: PSA_2D < 140 Å^2^ and AlogP98 < 5. ADMET: Absorption, distribution, metabolism, excretion, and toxicity, BBB: Blood–brain barrier.

**Table 1 molecules-28-02490-t001:** Structure-based pharmacophore model hypothesis with their features and score.

Pharmacophore Model	No of Features		Feature Set		Selectivity Score		ROC	
	CDK4/6	Aromatase	CDK4/6	Aromatase	CDK4/6	Aromatase	CDK4/6	Aromatase
Pharmacophore_01	6	5	AAAADHarom	AHHHH	1.993	4.764	0.775	0.555
Pharmacophore_02	6	5	AAAADH	AHHHH	1.993	4.764	0.909	0.546
Pharmacophore_03	6	4	AAAADH	AHHH	1.993	6.347	0.790	0.600
Pharmacophore_04	6	4	AAAADHarom	AHHH	1.993	6.347	0.643	0.570
Pharmacophore_05	6	4	AAAADH	AHHH	1.993	6.347	0.903	0.679
Pharmacophore_06	6	4	AAAADH	AHHH	1.993	6.347	0.654	0.704
Pharmacophore_07	6	4	AAAADHarom	AHHH	1.993	6.347	0.792	0.593
Pharmacophore_08	6	4	AAAADH	AHHH	1.993	6.347	0.904	0.582
Pharmacophore_09	6	4	AAAADH	AHHH	1.993	6.347	0.774	0.815
Pharmacophore_10	6	4	AAAADHarom	AHHH	1.993	6.347	0.625	0.715

**Table 2 molecules-28-02490-t002:** Fit value of identified candidates mapped to pharmacophore models.

Sl.no	Compound	Compound ID	Fit Value	
			Pharmacophore_02 Model	Pharmacophore_09 Model
1	Candidate 1	ZINC77287236	0.748	1.887
2	Candidate 2	CHEMBL517070	3.538	3.064
3	Candidate 3	51000421	2.642	3.515
4	Candidate 4	ZINC36924410	3.060	3.383

**Table 3 molecules-28-02490-t003:** Molecular docking result of identified candidates docked to target CDK6 and aromatase protein.

Sl.no	Compound	5L2S		3S7S	
		CDOCKER Energy (kcal/mol)	CDOCKER Interaction Energy (kcal/mol)	CDOCKER Energy (kcal/mol)	CDOCKER Interaction Energy (kcal/mol)
1	Candidate1	−13.04	−56.37	3.892	−56.27
2	Candidate2	−18.21	−46.60	−25.29	−52.21
3	Candidate3	−31.82	−35.28	−28.57	−32.98
4	Candidate4	−30.69	−32.07	−30.35	−31.16
5	Abemaciclib	−27.35	−58.02	-	-
6	Exemestane	-	-	−22.01	−39.15

**Table 4 molecules-28-02490-t004:** In silico ADMET profile and mutagenic prediction of the identified candidate compounds.

Compound	Absorption	Solubility	^a^ BBB	^b^ PPB	^c^ CYP2D6	Hepatotoxicity	AMES Mutagenicity	AlogP98	^d^ PSA-2D
Candidate 1	0	3	4	True	False	True	NM	1.034	123.8
Candidate 2	3	2	4	False	False	True	NM	0.434	185.8
Candidate 3	0	3	3	True	False	True	NM	0.552	82.35
Candidate 4	0	3	3	True	False	True	NM	0.175	82.35

^a^ Blood–brain barrier, ^b^ Plasma protein binding, ^c^ Cytochrome P450, ^d^ Polar surface area.

**Table 5 molecules-28-02490-t005:** HOMO and LUMO values calculated for identified candidates.

Sl.no	Compound	HUMO	LUMO	Energy Gap (ΔE)
1	Candidate 1	−0.128	−0.075	0.053
2	Candidate 2	−0.231	−0.094	0.137
3	Candidate 3	−0.192	−0.056	0.136
4	Candidate 4	−0.187	−0.061	0.126

**Table 6 molecules-28-02490-t006:** DS 2019 guidelines for ADMET-related descriptors and mutagenicity prediction.

ADMET Descriptor	Level	Description
Absorption	0	Good absorption
	1	Moderate absorption
	2	Low absorption
	3	Very low absorption
Solubility	0	Extremely low
	1	Very low, but possible
	2	Yes, low
	3	Yes, good
	4	Yes, optimal
	5	No, too soluble
	6	Unknown
BBB ^a^	0	Very high
	1	High
	2	Medium
	3	Low
	4	Undefined
	5	Unknown
PPB ^b^	0 (False)	Binding is <90%
	1 (True)	Binding is ≥90%
CYP2D6 ^c^	0 (False)	Non-inhibitor
	1 (True)	Inhibitor
Hepatotoxicity	0 (False)	Non-hepatotoxic
	1 (True)	Toxic
AMES Mutagenicity	0 (False)	Non-mutagen
	1 (True)	Mutagen

^a^ Blood–brain barrier, ^b^ Plasma protein binding, ^c^ Cytochrome P450.

## Data Availability

The data presented in this study are available on request from the corresponding author. The data are not publicly available due to privacy restrictions.

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
