# Peer review of "Virtual Screening for Identification of Dual Inhibitors against CDK4/6 and Aromatase Enzyme"

_molecules, 2023, doi:10.3390/molecules28062490_

Round 1

Reviewer 1 Report

In this article authors provide the rationale and describe their efforts aimed to find the chemical compounds inhibiting both aromatase and CDK4/6 in silico. According to the evidence described by the authors such dual inhibitors should be beneficial for treating HR+ breast cancer. Authors conducted several computational studies to find effective inhibitors in silico using modern software: pharmacophore-based virtual screening among compounds structurally similar to the known CDK4/6 and aromatase inhibitors, docking and molecular dynamic simulations. Also, they computationally assessed the ADMET and mutagenic properties for the four most promising compounds.

Comment:

1.      There is an article describing the sc-PDB, please cite it (I am not affiliated with the authors) https://pubmed.ncbi.nlm.nih.gov/25300483/

Author Response

Dear Reviewer

Thank you very much for your valuable comments and suggestions

The manuscript has been revised in response to your suggestions. Here are our responses to your comments and suggestions

Sincerely yours

On behalf of authors

Dr. Honnavalli Yogish Kumar

There is an article describing the sc-PDB, please cite it (I am not affiliated with the authors) https://pubmed.ncbi.nlm.nih.gov/25300483/

“We have cited the above references (Cite no 28)”

Reviewer 2 Report

This manuscript contains far to many grammatical and technical issues for publication.  The authors somewhat 'unethically' are stretching their data to describe hits/inhibitors when in reality they are potential candidates identified computationally. They were asked to fix this and did not.

In addition, the figures are very poorly made with some font sizes being well below what is visible to the reader.

Major revisions are needed many of which were not called out in the attached PDF, as it is not the reviewers job to proofread the manuscript for the authors.

This could be acceptable if:

• The authors properly describe their study and results as potential candidates and describe that they DID NOT screen anything 

• Remove all descriptions of the fact they discovered dual inhibitors or inhibitors at all.

• Fix the figures so all items can be read (no small fonts, blurry items, super thin lines)

• Proofread the manuscript carefully

• The authors provide a brief description of their candidates screened biological action but did not conduct any computational studies to explore what this would mean to their potential use as  CDK4/6. One would like to understand computationally what the established targets for each candidate would have and what structural information they can obtain from this work. For instance, what does the fact that candidate 2 was shown to target steroid sulfatase mean to this study?

Author Response

Dear Reviewer
Thank you very much for your valuable comments and suggestions
The manuscript has been revised in response to your suggestions. Here are our responses to your comments and suggestions
Sincerely yours
On behalf of authors
Dr. Honnavalli Yogish Kumar

This manuscript contains far to many grammatical and technical issues for publication.  The authors somewhat 'unethically' are stretching their data to describe hits/inhibitors when in reality they are potential candidates identified computationally. They were asked to fix this and did not.
“In this article, highlighted hits are computationally identified molecules that are known to bind to the target and binding studies was been evaluated by docking and molecular dynamic studies Therefore, can be termed as Virtual Hits or Hit candidates.”
In addition, the figures are very poorly made with some font sizes being well below what is visible to the reader.
“Corrected”
Major revisions are needed many of which were not called out in the attached PDF, as it is not the reviewers job to proofread the manuscript for the authors.
This could be acceptable if:
• The authors properly describe their study and results as potential candidates and describe that they DID NOT screen anything 
“We have well taken your advice and updated accordingly”
• Remove all descriptions of the fact they discovered dual inhibitors or inhibitors at all.
“Suggestions incorporated”
• Fix the figures so all items can be read (no small fonts, blurry items, super thin lines)
“Corrected”
• Proofread the manuscript carefully
• The authors provide a brief description of their candidate’s screened biological action but did not conduct any computational studies to explore what this would mean to their potential use as CDK4/6. One would like to understand computationally what the established targets for each candidate would have and what structural information they can obtain from this work. For instance, what does the fact that candidate 2 was shown to target steroid sulfatase mean to this study?
Sir, “From our end, we tried our best to fulfill your requirement to make this study publishable. Please find the answer to your query, especially under section 2.2 ” 

Reviewer 3 Report

Dear authors,

the work is really intriguing and well thought out, however in order to claim to have identified promising dual inhibitors, the identified compounds must be tested for the experimental validation. In my opinion it is necessary to complete the research with biological inhibition studies for aromatase and CDK4/6 prior to publication.

Author Response

Dear Reviewer
Thank you very much for your valuable comments and suggestions
The manuscript has been revised in response to your suggestions. Here are our responses to your comments and suggestions
Sincerely yours
On behalf of authors
Dr. Honnavalli Yogish Kumar

Dear authors,
the work is really intriguing and well thought out, however in order to claim to have identified promising dual inhibitors, the identified compounds must be tested for the experimental validation. In my opinion it is necessary to complete the research with biological inhibition studies for aromatase and CDK4/6 prior to publication.
Dear Sir, “We agree it needs to be tested and planned as a future study’’

Reviewer 4 Report

The manuscript entitled "Identification of promising dual inhibitors of CDK4/6 and aromatase: pharmacophore-based virtual screening, molecular docking, MD simulations, and ADMET analysis" reports a study using a structure-base in silico approach to identify potential compounds as inhibitors of CDK4/6 and aromatase. After pharmacophore models for each target, docking, molecular dynamics, and ADMET predictions the authors propose some candidates. The reported results can be used in further studies, however, there are some points to change and clarify.

1 - Abstract - page 1 - lines  47 and 48 

"...four potential candidates (candidate 1, 2, 3, and 4)..."

Please, how many compounds were screened? What database was screened? Please, add this information to the abstract. 

2 - Abstract  -  page 2 - 

"Further, ADMET mutagenic property predictions of all candidates (1, 3, and 4, Figure 6a) showed"

All candidates selected were 1, 2, 3, and 4, please clarify.

3 - Please, the introduction section must be shortened. 

4 - Please, improve the caption of table 1. 

5 - Please, figure 7 can be moved to the supplementary material. 

6 - Figures 8, 9, 10 and 11

Please, these figures can be merged. Please, I strongly suggest replacing the 3D  interactions plot with a 2 D interactions plot to show the similarity of the interactions. 

7 -  Results and discussion and conclusion.

Since there are no in vitro or in vivo tests. Please,  it will increase the visibility of the results if the authors compare the proposed inhibitors with some results of similar compounds reported in the literature.

Author Response

Dear Reviewer

Thank you very much for your valuable comments and suggestions

The manuscript has been revised in response to your suggestions. Here are our responses to your comments and suggestions

Sincerely yours

On behalf of authors

Dr. Honnavalli Yogish Kumar

The manuscript entitled "Identification of promising dual inhibitors of CDK4/6 and aromatase: pharmacophore-based virtual screening, molecular docking, MD simulations, and ADMET analysis" reports a study using a structure-base in silico approach to identify potential compounds as inhibitors of CDK4/6 and aromatase. After pharmacophore models for each target, docking, molecular dynamics, and ADMET predictions the authors propose some candidates. The reported results can be used in further studies, however, there are some points to change and clarify.

1 - Abstract - page 1 - lines 47 and 48

"...four potential candidates (candidate 1, 2, 3, and 4)..."

Please, how many compounds were screened? What database was screened? Please, add this information to the abstract.

“Corrected accordingly”

2 - Abstract  -  page 2 -

"Further, ADMET mutagenic property predictions of all candidates (1, 3, and 4, Figure 6a) showed"

All candidates selected were 1, 2, 3, and 4, please clarify.

Only selected four candidates were evaluated for expected mutagenicity using computational methods.

 “We have rephrased the above sentence in order to understand it clearly”

3 - Please, the introduction section must be shortened.

Dear sir,  We have tried to shorten the introduction.”

4 - Please, improve the caption of table 1.

“Corrected.”

5 - Please, figure 7 can be moved to the supplementary material.

“We have revised Figure 7 as suggested by others reviewer.”

6 - Figures 8, 9, 10 and 11

Please, these figures can be merged. Please, I strongly suggest replacing the 3D  interactions plot with a 2 D interactions plot to show the similarity of the interactions.

“Dear reviewer, we have well taken up your suggestions and updated accordingly, but we were unable to combine the mentioned figures. Although all the figures would be in one location, the interaction descriptions of each complex would be separated from their specific figures.”

7 -  Results and discussion and conclusion.

Since there are no in vitro or in vivo tests. Please, it will increase the visibility of the results if the authors compare the proposed inhibitors with some results of similar compounds reported in the literature.

“We have well taken up your valuable suggestions and updated accordingly. Please fine the same under the section 2.2 with references 32 and 33”

Round 2

Reviewer 2 Report

The authors have met the needs of my requests and now describe finding virtual candidates instead of hit molecules. They clearly state that experimental validation is needed.

That said, the manuscript is loaded with typographical errors and is not near a state of publication.  The authors must send a draft without spelling and grammatical corrections for further consideration. To be honest, the draft sent should have already had that.  It is not the reviewers job to proofread their manuscript.  

Author Response

RESPONSE TO REVIEWER-2 (round 2)

Dear Reviewer

Thank you very much for your valuable comments and suggestions

The manuscript has been revised in response to your suggestions. Here are our responses to your comments and suggestions

Sincerely yours

On behalf of authors

Dr. Honnavalli Yogish Kumar

Comment;

The authors have met the needs of my requests and now describe finding virtual candidates instead of hit molecules. They clearly state that experimental validation is needed.

That said, the manuscript is loaded with typographical errors and is not near a state of publication.  The authors must send a draft without spelling and grammatical corrections for further consideration. To be honest, the draft sent should have already had that.  It is not the reviewer's job to proofread their manuscript.  

Dear Sir “We have rectified the shortcomings that have been pointed out”

Reviewer 3 Report

Dear authors,

the biological validation is essential for this kind of publication.

Nonetheless, given the substantial amount of work completed, I hope you will be able to provide the experimental data as soon as possible.

Author Response

RESPONSE TO REVIEWER-3 (round 2)

Dear Reviewer

Thank you very much for your valuable comments and suggestions

The manuscript has been revised in response to your suggestions. Here are our responses to your comments and suggestions

Sincerely yours

On behalf of authors

Dr. Honnavalli Yogish Kumar

Comment;

Dear authors,

The biological validation is essential for this kind of publication. Nonetheless, given the substantial amount of work completed, I hope you will be able to provide the experimental data as soon as possible.

Dear sir “Thank you for your valuable feedback. We too, wish to complete the ongoing wet lab experiments as soon as practicable so we'll be able to report and publish the findings.”
